# Nurturing Leaders in Community-Based, Primary Healthcare Services for People with Disabilities in Low- and Middle-Income Countries

**DOI:** 10.3390/ijerph22040622

**Published:** 2025-04-16

**Authors:** Roy McConkey

**Affiliations:** Institute of Nursing and Health Research, Ulster University, Belfast BT15 1ED, UK; r.mcconkey@ulster.ac.uk

**Keywords:** leaders, leadership, community-based, primary care, disabilities, intellectual disability, low- and middle-income countries, trans-national, artificial intelligence

## Abstract

The health and social care needs of children and adults with disabilities are often neglected in many low- and middle-income countries. International opinion favours the creation of community-based supports rather than the institutional and clinic-based care that has dominated to date. However, models of care that are reliant on community leadership have been slow to develop within and across less affluent countries. Moreover, the managerial models inherent in institutional-based care are likely to be inadequate in such settings. This descriptive study aimed to explore the leadership qualities required in initiating and sustaining community-based supports. Face-to-face interviews were conducted with a purposeful sample of 16 leaders of projects in Africa, Asia, and South America. They included people with sensorial, physical, and intellectual disabilities as well as non-disabled leaders of local and national projects plus others whose leadership was at a regional or international level. Two main questions were addressed: what are the qualities required to function as a community leader and how can these qualities be nurtured in low resourced settings? The insights gained would inform the preparation and training of community leaders. Thematic content analysis identified three core themes: first, personal qualities such as empathy with an understanding of the personal circumstances of persons in need of support; second, communicating clearly the vision and values informing their work; and thirdly, building and mobilising community support from families and neighbours. The nurturing of leadership comes through mentoring and coaching, the empowerment of others, networking opportunities, and the development of inter-personal and communication skills. These themes were commonly expressed across the 16 leaders from all the participating nations and at all levels of responsibility, which suggests a universality of approach in relation to people with disabilities. The findings are in marked contrast to current practices in health and social care that have valued professional expertise over lived experience, knowledge, and technical skills over compassion and empathy, and the provision of person-centred “treatments” over developing community and personal self-reliance. Nonetheless, the challenges involved in establishing and sustaining new styles of leadership are many and will not be quickly resolved.

## 1. Background

The World Health Organization, in association with the World Bank, estimated that in 2021, over 4.5 billion people globally were not fully covered by essential health services [1]. The following are two further points of note: across the nations, the proportion of the population not covered varied from 14% to 87% with the higher proportions in low- and middle-income countries (LMICs), and from 2015 there had been minimal increase globally in the proportions of the population accessing essential services. Furthermore, the link between poverty and a lack of access to health services is well documented. Among the poorest in many countries are people with disabilities. According to the World Report on Disability [2], an estimated 50% of people with disabilities in many LMICs do not receive the healthcare they need, due to financial constraints, lack of accessible facilities, and inadequate healthcare services tailored to their needs [3].

Internationally, there is broad agreement that the manifold needs of people with disabilities are best met through community-based supports [4]. Likewise, universal coverage of the basic public health needs of the population has to start with families and local communities supported by primary healthcare services [5]. However, in less affluent nations, much of the healthcare budget is spent on high-cost clinic- or hospital-based services with insufficient trained professionals to address the health needs from birth through to old age [6]. Consequently, in low- and middle-income countries, infant and mother mortality is higher and overall life expectancy is lower [7] and quality of life is much less than in more affluent countries [8]. Moreover, these scenarios are forecast to persist up to 2050 unless targeted actions are taken [9].

The 17 Sustainable Development Goals (SDQs) [10] agreed by nearly all world’s nations aimed to redress the inequalities that are present internationally among, but also within countries. The aspirations of no poverty (Goal 1), zero hunger (Goal 2), and “healthy lives and promote the wellbeing for all at all ages” (Goal 3) were to the fore, yet it is very unlikely that these SDQs (and the other 14 goals) will be met by the target year of 2030 [10].

Among the most marginalised in every country when it comes to alleviating poverty and accessing healthcare are people with disabilities, especially those born with or who acquire intellectual and developmental impairments [11]. In addition to their physical health needs, they and their families often suffer from poverty, poor housing, and inadequate sanitation, allied with social exclusion from education and employment opportunities due to the stigma attached to their disability [12]. These inequalities can be present even in high income countries especially in families facing multiple disadvantages, albeit not to the same extent than their counterparts in poorer countries [13].

The arguments for greater investment in primary healthcare and in community rehabilitation for disabled persons have been well rehearsed, although their implementation remains disappointing, especially for marginalised groups [14]. Many of the innovative projects depended on international donors and were not sustained when the funding finished. More significant is the lack of professional expertise—doctors, nurses, and therapists—to sustain these new styles of home- and community-based supports. The clinicians that were available in-country tended to be confined to hospitals or clinics nor did they have the training and experience in these new care models so that they could mentor and support community personnel. Hence, across all nations, the challenge remains as to how new forms of family-centred, community-based supports can be provided to people with intellectual disabilities, in particular, when human and financial resources are constrained [15].

In recent years, attention has shifted as to how indigenous resources could be mobilised to address the unmet needs in primary healthcare through community leadership and engagement [16] and the provision of support for family caregivers of people with intellectual disabilities [17]. For example, people with the lived experience of disability and their allies, such as family members and local persons of good will, could have the insights and knowledge to devise and implement new forms of support [18]. Recent literature has provided valuable insights into the form these new services can take, along with examples of the diversity of responses that were devised for impoverished communities. A key element to success was the leadership of these community initiatives [19].

Past research in health and social care has tended to focus more on managerial models of leadership that are commonly found in national health services provided by government agencies, even at a local level. International and national Non-Governmental Organisations (NGOs) have often emulated them also [20]. These models tend to focus on managers having supervisory, budget management, and accountability functions in hierarchical structures with the managers at the top, often removed from personal contact with service beneficiaries and also with the front-line workers with whom their “patients” have contact. In such systems, there are usually defined procedures to be followed by workers at each level of the managerial structure [21]. By contrast, community services need to operate on a more fluid, personalised model that can respond to the varied needs of the beneficiaries that transcend traditional service boundaries of health, education, and social welfare, for example. Moreover, the values underpinning these models often differ. This had led to a call for a re-appraisal of leadership in public health and primary care towards a more adaptive style of leadership and not of a traditional “command and control” variety [22].

A recent report—covering LMICs in Asia, Africa, and Latin America—focused on what was broadly termed “character-based” leadership; typified by moral purpose and the personal qualities of leaders [23]. The literature review of over 1000 published articles identified a marked increase into these leadership styles especially from 2020. The majority of studies were within the corporate section and around one quarter in education but only 10% in healthcare and 2% in the not-for-profit sector. Over 70% examined the impact of this form of leadership using mostly quantitative methods with just over 10% describing the characteristics of leaders of which the virtue of humility was the most studied. Other valued traits such as empathy, resilience, trust, and courage were rarely investigated. Moreover, only 15% of over 700 leadership training programmes examined had a clear focus on character leadership, and of these, the majority were delivered by NGOs and philanthropic foundations followed by universities.

Servant leadership was the most commonly named approach in this report with nearly one quarter of articles devoted to it along with a variety of other leadership styles. Another systematic review of nearly 300 published studies on servant leadership [24] brought greater clarity to the core components of this style of leadership, which were motive (being of service to others, not self), mode (led by the individual needs of others), and mindset (concern for the larger community and a commitment to be accountable for their well-being). The authors noted that the dearth of field research is an impediment to better understanding—for example—as to how servant leaders could create more servant leaders using a “trickle-down approach” based on a social learning process [25]. Nonetheless, the authors caution that tremendous efforts are needed to develop a servant leadership culture within organisations and maintain its spread throughout larger systems.

To date, there has been a paucity of research on effective leadership models for community-based services addressing the needs of people with disabilities. By contrast, a number of studies have examined leadership in community mental health services albeit within high income countries. For example, an evaluation of 16 community projects in the USA for improving the mental well-being of men and boys identified five leadership qualities that contributed to success, namely personal vision, value-based leadership, relationship- and task-oriented leadership, and opportunities for leadership development [26].

Despite the increased interest in character-led, leadership styles, especially in LMICs, there has been limited impact to date on public health and community care systems. Few studies have been undertaken into the qualities required by leaders of community-based supports for people with disabilities, especially ones that are instigated, managed, and maintained by local personnel, most of whom will do so on a voluntary basis and within a culture of empowerment and self-reliance [27].

### The Aims of the Study

The present study aimed to address this gap. Informants were purposively selected from those who were currently engaged in community-based initiatives to assist youth and adults with intellectual disabilities in mainly low- and middle-income countries at various levels, namely at a locality, national, and a trans-national, regional level. Two questions were posed: 1. What are perceived to be the personal qualities and skills needed for a successful leader in local, community-based services? 2. What has proved successful in nurturing and supporting people in a leadership role? The insights gained in this small-scale, descriptive study could contribute nevertheless to the training and development of future leaders who will be needed in order to expand the opportunities for more communities within a country to develop their own locality programmes.

## 2. Method

A qualitative methodology informed by interpretative phenomenology [28] was chosen to gain insights based on the lived experience of respondents who were currently involved in leadership roles in community settings, as well as having prior experience in other roles. Face-to-face interviews were used as they promised richer insights than self-completion questionnaires. During the interview, various probes could be used to encourage respondents to expand or clarify their comments. Purposive sampling was selected with the aim of recruiting a diverse sample of respondents balanced by gender and scope of the leadership, including people with disabilities or their advocates from a range of countries mostly in the global south and whose role had national and regional responsibilities as well as a local focus. Recruitment continued until data saturation in terms of recurring themes was attained.

To be clear, the insights gained from this sample were not intended to be representative of current leadership in community-based services or to evaluate the extent to which the leadership qualities were currently being implemented and their impact on communities. Rather the aim was to scope the features of leadership based on the lived experience of a diversity of leaders and from which a conceptual framework would emerge. Such a framework could be used, for example, in the recruitment of future leaders and to aid the self-reflection of persons in leadership roles as to their personal development needs.

### 2.1. Procedure

Formal ethical approval was not sought for the study nor was it required as it met the criteria in the United Kingdom for an evaluation of participants’ service provision rather than a research study. Nonetheless, ethical considerations from the World Medical Association, Declaration of Helsinki on research involving human subjects [29] were fully observed, namely respondents were given details of the study, the information they would be asked to provide, the purposes to which the information would be used, and they could withdraw their consent at any stage without giving a reason. They were also asked to give explicit consent to the interview being audio-recorded.

A snow-balling approach was used to recruit participants during their participation in an international conference convened in the United Arab Emirates on inclusion in education, sports, and employment for persons with primarily intellectual disabilities in October 2024. These participants were supplemented by two, in-country interviews in the same month. All the people approached agreed to take part. Table 1 describes their characteristics.

The author personally conducted the interviews in English as chosen by the interviewees. They were mostly carried out face-to-face in a private setting but one was carried out via Zoom. The interview had three parts: First, the interviewees briefly summarised their leadership roles and backgrounds. Second, they were then asked to describe the qualities and skills required of leaders based on their experience, including what they personally contributed to this role. Third, they shared their thoughts as to how leadership could be nurtured in the communities they are working in or have worked in. The interviewer used probes throughout the second and third parts to elicit further elaboration or clarifications. These were chosen based on active listening, his knowledge of the literature, and personal experiences. The interviews lasted on average 10 min and generated nearly three hours of recordings. It was evident by the 14th interview that no new themes had been mentioned but 2 interviews that had been pre-arranged were conducted as a further check that data saturation was achieved.

### 2.2. Qualitative Analysis Using AI

The advent of new forms of Artificial Intelligence (AI) promises to transform healthcare through telemedicine, for example [30]. Thus far, its deployment within health research has been limited yet examples are emerging of its use especially in qualitative research [31]. Nonetheless, human researchers are urged to use AI as an assistant rather than a replacement: the so-called hybrid approach [32]. This study provided an opportunity to explore the use of a hybrid approach when analysing the interviews using AI, in this instance a freely available tool: Microsoft Co-pilot.

Two advantages of AI are first, the speed and efficiency offered by AI as it can identify the themes in each interview in less than 30 s. In turn, AI can go on to quickly identify the themes that occurred most often across all the interviews. Consistency is a second advantage as the AI algorithms apply the same criteria uniformly, reducing variability and ensuring consistent analysis for each interview. By contrast, a human analyst could be biased in their analysis of interviews given by certain persons and it would also take very much longer.

AI has its weaknesses though, three of which are particularly relevant to this study. The author brought an understanding of the contexts in which the interviewees worked, and he could interpret nuances, context, and subtleties in their responses that AI would miss due to its reliance on the written words. Secondly, as he had personally conducted the interviews, he could empathise and understand emotional cues, providing a more personalised and intuitive analysis. Thirdly, there was the risk that informants who provided a longer and more detailed interview could bias the outcomes compared to those who had more limited communication skills, either as a result of a disability or using English as a second language. Thus, it was essential that member checking was undertaken with all the interviewees so that they too were part of the human contribution to validating the initial analysis by AI.

The initial transcript from the audio recording was made using TurboScribe (AI), a Web-based, free service. The Word file was examined by the interviewer for clarity and correction. A copy was also sent to each interviewee for checking and changing. For persons with visual impairments, the audio-recording was sent although both had access to screen readers. Few corrections or changes were made by the interviewees, which broadly reflected the accuracy of the AI transcription.

### 2.3. Data Analysis

The analysis of the interviews, although informed by interpretative phenomenology analysis, was largely based on thematic content analysis using Microsoft Co-Pilot to initially code the themes in each transcript inductively. In brief, this is how Co-Pilot describes how it works: “Copilot uses a combination of natural language processing (NLP) techniques and machine learning algorithms”. This includes breaking the text into smaller units, such as words or phrases; using algorithms like Latent Dirichlet Allocation (LDA), or clustering techniques are used to identify recurring themes or topics within the text; analysing the emotional tone of the text to understand the sentiments expressed, extracting important keywords and phrases to highlight significant points, and then the identified themes are reviewed and refined to ensure they accurately represent the content of the interview. This replicates much of the recommended practice for undertaking a thematic content analysis [33].

An anonymised version of each transcript was uploaded to Co-Pilot and the following question was posed: What are the main themes relating to leadership in this interview? The response was saved to a Word file. The themes identified in each interview were checked by the author on the interview transcript on a line-by-line basis. Particular attention was paid to confirming and ensuring that the themes identified by AI were present in the total corpus of interviews but also other themes that had not been captured by AI. These were few in number and usually concerned issues that were tangential to leadership. In addition, the author identified quotes from respondents that captured the essence of each theme as AI had given limited and in some instance inappropriate examples.

Co-pilot was again used to synthesis all the themes identified in the 16 interviews and list them into those which were commonly mentioned: in total, 20 were identified. However, further attempts to have AI identify the inter-relationships among the themes did not yield consistent outcomes: for example, 11 main themes were returned. Instead, the author re-examined the transcripts and the listed themes, and he identified three core themes and the subthemes within each, following the recommended procedures for thematic content analysis [33]. This analysis was informed by his personal experience of interviewing the informants, his previous engagement with community-based programmes, and his past use of interpretive phenomenological analysis in order to reflect the coherency around the lived experience across these 16 informants [34].

These summaries were then sent to the 16 interviewees for member checking as they were deemed best placed to assess the validity of the analysis. No disagreements were reported with the core themes and the subthemes, and no additional themes were proposed although some persons commented on what they considered to be themes of particular importance, and some adjustments were made in light of their feedback.

Additionally, Co-pilot was asked to undertake a deductive coding of this final analysis and the style of leadership that it represented, which was refined with reference to the extant literature that Co-pilot was requested to source.

## 3. Findings

### 3.1. Sample

Table 1 summarises the characteristics of the sample of people who participated in the study: 16 in all. Six were based in Sub-Sharan Africa, three each from Asia and Latin America, two from the Middle East, and one from Eurasia, plus a further person who had a global remit. Five had a sensorial, physical, or intellectual impairment and one was a parent of a child with a disability. There were equal numbers of males and females. Details of ages were not collected but most were in their 20s or 30s. Information about their various roles and backgrounds are given in the table.

### 3.2. The Core Themes and Subthemes

Figure 1 presents the three core themes relating to leadership that emerged from these analyses. The central theme combined the personal qualities of leaders which guided their communications with others and in efforts to build communities. In describing the core themes, each constituent subtheme is briefly described but supported by direct quotations from interviewees to ensure their voices took precedence (see Table 1 to identify the interviewee from the assigned code number).

#### 3.2.1. Personal Qualities of Leaders

Various qualities were noted and although they are reported separately, there is an overlap across them and ideally a coherence among them that makes their leadership especially effective. The following subthemes emerged from the analysis.

Empathy was a fundamental quality across the interviewees, gained through active listening to those whom the leader is seeking to help.


*A leader who listens to a voice that comes from the other people that you maybe leading and listening to them and taking actions towards those things. It shows braveness in you and shows kindness in you. And it also builds someone to be a role model of that person (04).*



*The first thing is to be able to listen to what people are saying. Especially when you think about persons with disabilities, because they have a lot of grievances within themselves. Never be … too hard on them because, you see, the issue is that these people are already broken (02).*



*I think that the most important quality for leadership is empathy. You have to put your shoes in the others’ shoes … that’s really important (12).*


Leaders needed to have an inclusive mindset in that they not only welcome others, but they seek to engage them and value their contributions.


*Leadership should be inclusive, and invite all people to the table. Not just to sit at the table, but to be a part of the discussion, or a part of the action (16).*



*I think that the first and most important priority is to find someone who is a bridge builder, because essentially creating social inclusion means bringing together sometimes services, sometimes communities, sometimes neighbourhoods or groups of people or families who would not naturally find each other, or perhaps find communities of each other even if they were in the same physical space (10).*


Self-confidence allied with humility was another facet of leadership.


*Be confident of who you are. Be confident in what you do. Be confident when you speak. Be confident to fight for your rights (04).*



*You should have the humility to learn from other people (13).*



*A humble leader then is putting other people forward, is celebrating their work. It’s not taking all of the credit for themselves (14).*


Leaders appreciate and mobilise the talents of the people they lead.


*I see a lot of qualities in many people, actually in all people. And it’s interesting because every person have like a different quality. Together it is like a (Jigsaw) puzzle inside all the group (12).*



*A leader who listens to a voice that comes from other people that you are leading and listening to them and taking action towards those things (04).*



*We work greatly with the self-respect with each person and to put value on what we are, what we have, and not in what we don’t have (11).*


Leaders need to be resilient through the challenges and difficulties they will encounter.


*Be someone who can keep the lines of communication open through thick and thin, accept criticism and be able to take accountability when needed (05).*



*I persevered when there were troubles, because, I mean, organisations sometimes can find themselves in trouble. But we persevered together, and we endured together, we moved on. There were a lot of difficult times for the organisation. But they were able to see that I motivated them, I encouraged them. And they also encouraged others (02).*



*What I would tell my fellow colleagues with intellectual disabilities is never give up. Never feel that you’re not home. Be confident of who you are. Be confident in what you do. Be confident when you speak. Be confident to fight for your rights (04).*



*The self-confidence to be truly vulnerable, to admit when you don’t know something, to ask for help, to seek out advice, because it makes others know that they are truly participating (10).*


Other qualities were also mentioned by a few individuals: patience, commitment, self-reflection, curiosity, sense of fun, and personal development and growth.

#### 3.2.2. Communication

The second core aspect of leadership was around communicating the vision and values that should guide their endeavours, how to create team-working, and finding creative ways of problem-solving.

Vision setting and creating a shared spirit was perceived to be a key function for leaders.


*Spirit comes first before resources. If we have a spirit, we can collect resources, mobilise resources, but if we have resources but no spirit, then that leadership doesn’t work out (01).*



*A leader does not mean that you have to just know how to read or write, but a leader should just know what to do and stand up for our rights. In Africa, it’s not easy to get a job. It’s not easy to have an education system. But we’re trying to find them, to advocate for ourselves to get meaningful roles, jobs, even education systems (04).*



*When I speak about communication, it’s not a complicated purpose or technologies, because in the community, it’s people based. So how you talk to people, how you make them feel, how you convince them and how you make them feel to own that vision (13).*


The values inherent in the vision—such as inclusion and rights—also need to be emphasised and shared.


*Inclusion for us is such a kind of world where people living with and without disabilities can work together, can march together, can talk together, can laugh together, can cultivate friendship, and ultimately they can go and grow together at the same time (01).*



*Making inclusion an inherent value and … meet up, discuss and maybe share the messages that you want to convey or that you want people to hear about and try to take those messages in ways that can be accessible for people of all abilities (05).*


Communicating and promoting teamwork among the community is another important role of leaders.


*You have to cultivate your connections with the people you work with because these are the ones who will be going to work things out for your vision. The community develops you as a leader and then you develop community (09).*



*You need to identify yourself as part of a community. You need to be taking active part of any actions that is meant to be destined for the well-being of the community and for you (15).*



*A leader that’s able to make people come together to feel there’s a safe space to raise their concerns, to offer ideas. They have to get their hands dirty and be at the coalface of that work (14).*


The leader also needs to be adept at facilitating finding solutions to the problems the community faces.


*Figuring out how you can do the best you can with what you have, and how you can creatively sort of exploit or see opportunities where others might not, is absolutely essential (10).*



*Leadership is something, one could practice to be calculated risk takers, not kind of comfort seekers (01).*



*One of the important assets to possess as a leader is the ability to be flexible and adaptive, which I believe people with disabilities have mastered (05).*


#### 3.2.3. Building Community

The third core theme in leadership was that of forging relationships and the building of community. The following subthemes were mentioned.

Having personal knowledge and experience of the community is essential.


*It’s important for you to understand the DNA, the social fabric of that community, what makes them tick … the dynamics of that community (10).*



*You need to identify yourself as part of a community. You need to be taking active part of any actions that is meant to be destined for the well-being of the community and for you (15).*


A focus on identifying and utilising peoples’ assets within their community is also advisable.


*If you want to work to be great, work with those people who are in the local level, because they are the people who want the work to be done (06).*



*Creating social inclusion means bringing together sometimes services, sometimes communities, sometimes neighbourhoods or groups of people or families who would not naturally find each other (10).*


Consultation with communities is vital to identifying their needs and priorities, and how best they can be met.


*It needs you to be open to new ideas, to new ways of thinking, and also to gather all those ideas and to process everything with your own ideas, your own way of being as a person. At the end you would be building community, not only for you, but also for the rest of the people that are part of the work that you are doing (15).*


Leaders should mobilise others to participate in making change happen.


*A good leader also has to be a good follower … you are not leading a system or process, you are leading them through people (09).*



*I think we can focus much more deeply on participatory forms of development and research to make sure that when we are going into a community and we’re talking about social inclusion, we have some ideas, we have some tools, but we don’t have a prescriptive process. We don’t just sort of hand over an instruction manual and say, this is how you do it (10).*


#### 3.2.4. Styles of Leadership

Many styles of leadership have been described in the literature [21]. A comparison was then made to match the foregoing description of leadership as perceived by these interviewees with those previously described in the literature. Initially, Co-pilot was used to do this by uploading the foregoing analysis and asking it to identify the leadership styles that the framework portrayed. As noted below, three leadership styles were named and described which were cross-checked by the author from the transcripts and subsequently by sharing them with the interviewees.

Transformative leadership aims to inspire communities to work towards a common vision through innovative actions that encourage creativity and problem-solving and find new ways to support the community. This is illustrated in a comment from one interviewee.


*The best leadership for me is what I call transformative leadership. It’s leadership that brings change and leadership that brings change is not the same as leadership that is for managing day-to-day tasks” (13).*


Democratic leadership fosters inclusive decision-making by encourages participation and input from all members. It promotes teamwork, collaboration, and shared responsibility, which can be beneficial in a resource-constrained environment. This is illustrated in this comment about people with disabilities in leadership roles.


*We should give, not just for leadership’s sake, but we should give them some sort of power to exercise so that they could exhibit their real abilities no matter whatever their physical limitations are (01).*


Servant leadership prioritises the needs and well-being of the community and emphasises empathy, listening, and caring for others, which are crucial in understanding and addressing the unique needs of people with disabilities. As one interviewee commented using the term “compassionate”:


*It’s a leader that listens to others. Compassion is the mainstay of what it would be, because if it’s not a compassionate leadership, then it’s not going to rally many people for the long term” (14).*


Co-pilot identified servant leadership to be the best fit overall, stating, “It encompasses a majority of the qualities you identified, particularly those related to empathy, community-building, and inclusive, participatory approaches”. It added, “This model is well-suited for community-based, not-for-profit organizations focused on helping people with disabilities, as it prioritizes serving others and fostering a supportive, collaborative environment”.

However, the author felt this under-estimated the transformative leadership ambitions that were inherent in the passion and commitment of the interviewees to make changes happen. Hence, a fusion of transformative and servant leadership would be a better description. But perhaps no single label encompasses the style of leadership that these interviewees described, and elements of each style are likely to feature depending on the differing needs of the beneficiaries, the community context, and the focus of their efforts. The Discussion Section expands this point.

#### 3.2.5. Nurturing Leadership

The interviewees shared their views on how leadership could be nurtured within communities. They alluded to the view that leaders are born rather than created but in their experience, they felt anyone could become a leader.


*We all have the capacity and the ability to become a leader. The only thing is that you need to identify what is the cause that motivates you, that pulls those triggers inside of you that you say, okay, this is what I was looking for. This is something that really motivates me to do not even better, to do the best that I can (15).*


Many stressed that one-to-one mentorship and coaching is their favoured approach to developing leadership talent.


*There was need to nurture another crop of leaders. So one thing that I decided to do was to identify those people that really showed interest in the organisation. And then apart from identifying them, I put them quite close to me so that they could learn the qualities of leading that organisation” (02).*


Effective leaders believe in the potential of others and empower them. This involves recognising the unique contributions each person can make and trusting them to work on the chosen tasks, even at the risk of mistakes being made.


*People can make mistakes because we are not gods, but we should make them learn from their mistakes. We cannot do anything about the past, but the only thing we can do is learn from that mistakes and help them nurture (01).*


Networking with other leaders is also commended as a means for learning from their peers and boosting their motivation.


*I think a way is creating opportunities so they can share with people that feels the same as them. Learn good practices from the people around them (11).*


Acquiring knowledge through training opportunities can contribute to their development but in low- and middle-income countries, the lack of suitable trainers is an issue. Although, the greater benefit is likely to come from meeting and interacting with their peers.


*Coming to forums where we meet people from other areas also gives (leaders) a purpose for life in a way to say, okay, my fellows are doing it. Why not? Why can’t I? (08)*


Perhaps IT will open up new possibilities such as chat rooms and online, interactive workshops, as this mentor of a leader with intellectual disabilities notes.


*(Name) likes IT a lot that, you know, on all these platforms, he meets Google people. He’s inquisitive about it. Okay, how can I also use my IT to help me and also help other athletes? (08)*


## 4. Discussion

This study had a number of strengths. It focused on promoting the health and social well-being of persons with disabilities, which to date has received scant attention in terms of leadership research. Moreover, the purposive sample brought together informants from three continents with leadership experience at different levels within community-centred services and varied personal backgrounds. Nonetheless, there was a striking coherency in their experiences and views on leadership. The qualitative methodology opened up their lived experience to examination while the overall analysis process gave them an active part in validating the conceptual model that emerged. Furthermore, the use of AI demonstrated its potential for supporting future research and evaluation projects, especially those involving qualitative data analysis. That said, there were some significant limitations that will be discussed later and which further research might address.

### 4.1. Leadership Qualities

The conceptual model of leadership that emerged resonates with many of the features described in recent reviews of nearly 1300 studies undertaken in character-based and servant leadership [23,24]. Moreover, they reflect the leadership qualities reported in a study of community-based mental health supports in the USA [27] such as personal vision, value-based, and relationship-oriented leadership.

Perhaps it is not surprising that the same human qualities can be deployed in different fields of endeavour, such as business, education, sports, or healthcare, as well as across continents, because they all involve people assisting other people, be it their work colleagues or the people they serve and support. Also the associated leadership domains of communication and building communities are just as transferable across different contexts. Nonetheless, it could be that certain qualities within these core themes are more significant for leadership in particular settings. As our interviewees had a common focus on disability, the features they prioritised in their leadership were empathy, promoting a vision of inclusion, and being rooted in the communities they were seeking to support. Further comparative research across different beneficiaries of public health services, for example, maternal health or care of the elderly, might reveal a different set of leadership priorities.

Or maybe not. One of the features of leadership that our informants implicitly practiced was a focus on person-centred working, but which did not come across strongly in their interviews, and especially the need to be flexible and adaptable to different people and contexts. In retrospect, this feature of leadership would be worthy of deeper investigation given the heterogeneity among persons who are labelled as “disabled” or within a community of persons from different cultures. Thus, the qualities identified in any leadership framework are better viewed perhaps as a “tool-kit” from which leaders can select the combinations most suited to the people and communities they aim to support and the changes they are working towards. In that respect, the WHO framework on community-based rehabilitation is a valuable guide to different activities and outcomes [4].

The same argument applies to the value of trying to distinguish different leadership styles of which many have been identified, although the lack of studies into leadership within community-based health and social care is concerning [27]. A case can be made for identifying particular styles of leadership in terms of conceptual clarity and notably in comparing outcomes from different styles of leadership as some have done [24]. Yet in this study, our interviewees reflected elements from different styles of leadership without categorising them and when some did so, they came up with different labels. Hence, the main value of defining differences in leadership styles is arguably to widen the tool-box of strategies that leaders can use rather than prescribing one style as preferable to another. Nonetheless, some leaders may feel more comfortable with one style over others, which would further reinforce the need for leadership teams rather than depending on one person’s leadership style [35].

### 4.2. Nurturing and Developing Leaders

The main rationale for the study was to gain insights as to how leaders could be nurtured and developed. Surprisingly, the interviewees made little mention of providing formal training, which tends to dominate in the current literature, with universities and consultants seen as the main providers of leadership education [23]. In part, this may represent the unavailability of such options in the settings in which the interviewees work but a deeper reason could be the value they placed on the personal development of leaders through mentorship, coaching, role modelling, empowerment, and networking. All of which stem from the reasonable assumption that future leaders are in the communities with which they are currently engaged. Nonetheless, this may be overly optimistic as certain personality traits—such as humility and compassion—are not easily taught [24]. Consequently, wise leaders might also need to engage in some form of selection for potential future leaders based on their personal qualities and prioritise them over qualifications and experience.

The question remains as to how the conceptual framework described here, and similar ones identified in the associated literature [23], could be extended to existing primary care services or health and social care services for persons with disabilities. A strong case was made for this to happen in order to make these supports more effective and efficient through mobilising community initiatives [36,37]. A favoured approach has been the formation of community health committees [38] and the creation of new staff positions such as community health/rehabilitation workers [39,40]. Although these roles are intended to embody many of the attributes of leadership described in this and other studies, they need to be complemented by their service managers and colleagues also exhibiting similar styles of working within the extant health systems [41]. However, transforming existing systems and practices can prove more challenging and strategies for changing leadership styles and cultures, in particular, remain to be tried and tested [42].

### 4.3. AI as a Tool in Research and Evaluation

The use of AI to assist with the qualitative analysis of interviews was a novel experience for the author and the interviewees. It certainly speeded up the initial stages of the inductive coding of themes within each interview and across interviews, but AI seemed less successful in combining themes into a conceptual framework although it did help to confirm the links with different leadership styles. In part, this may be a reflection of the author’s inexperience in using AI plus the possible limitations of the chosen AI system: Co-pilot. The latter was chosen on the grounds of being freely available so that researchers and practitioners in LMICs could avail of it. Other AI systems may be better suited for qualitative data analysis although the evidence for this has yet to emerge and often the choice may be based on availability and accessibility. It is likely too that advances will be made in fine-tuning AI to undertake more accurate analyses [43]. In addition, we also used AI to assist with the literature searches (such as Google Scholar) and the writing of reports, for example, the translation into other languages (in this instance, Spanish) and producing a more accessible version of the study for a lay audience. Once again though, human cross-checking was undertaken to improve the accuracy and relevance of AI-generated content and overcome any biases.

Another reason for using AI was to explore its potential to empower and assist leaders in community-based services to undertake their own research and evaluation studies. Often, they have limited or no access to additional funding to do this, yet their experiences, if documented, are vital to improving the reach and relevance of public health efforts notably in low resource settings and less affluent countries [36]. Nonetheless, concerns around governance and ethics in the use of AI will need to be addressed [44] alongside the development of training opportunities, including the use of AI, as an aid to encouraging service evaluations and reporting [45].

### 4.4. Limitations and Future Research

Finally, the limitations of this study need to be noted. The sample was biased to leaders who broadly promoted health and well-being of persons with disabilities rather than primary care or public health staff involved in delivery of disease management and prevention. As noted above, this is a vital area for further research. Also the views of leaders need to be complemented by documenting their impact on beneficiaries and the communities’ perceptions of the leadership attributes that they valued [35], which we were unable to do due to limited resources. Longitudinal studies with larger samples of informants would be valuable although the costs involved in such research can be prohibitive. In addition, more in-depth studies into the cultural influences on leadership styles and the impact they have on the health and well-being of beneficiaries are needed [46] but such trans-national studies are costly and challenging to arrange and manage [47]. Hence, mobilising local personal to undertake their own evaluation could be one way forward. Indeed, this small-scale study is a modest example of the contribution that AI could make to health and social care research, but more extensive evidence is needed in terms of its added value and how its limitations and risks can be addressed particularly in evaluating the impact of leadership on service outcomes.

## 5. Conclusions

Leadership in community-based primary care and disability services was perceived to involve three core themes: personal qualities such as empathy, communicating a clear vision and values, and building and mobilising community support from families and neighbours. According to these informants, this style of leadership is best developed mainly through mentoring and coaching, the empowerment of others, networking opportunities, and the development of inter-personal and communication skills. These themes were commonly expressed across the 16 leaders from all nations and at all levels, which suggests a universality of the approach in relation to people with disabilities.

These findings are in marked contrast to practices in health and social care services in many countries that have tended to value professional expertise over lived experience, knowledge, and technical skills over compassion and empathy, and the prescription of “treatments” to individuals over community and personal self-reliance. Nonetheless, the challenges involved in establishing and sustaining new styles of leadership are many and will not be quickly resolved. But naming them is a first step to their removal, assisted by concerted efforts to foster and finance new styles of community-based leadership.

## Figures and Tables

**Figure 1 ijerph-22-00622-f001:**
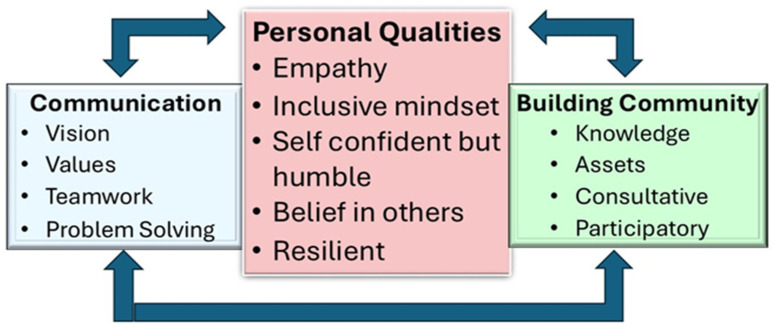
A conceptual framework of community leadership.

**Table 1 ijerph-22-00622-t001:** A description of the participants (*n* = 16).

Code	Role	Country/ Region	Personal Statement	Status	Gender
01	Co-founder of a youth-led, volunteer-based organisation called Rights as a Society.	Nepal	I have a visual impairment and have graduated from University. I’m passionate about leadership entrepreneurship in grassroots movements, so that people with disabilities would be able to speak up for their rights and also they would be able to stand on their own feet.	Country National	Male
02	A disability activist and Disability Officer with the national office of an international NGO.	Zambia	Blind from when I was 10 years old, I graduated from University with a PhD and previously worked as a university lecturer.	Country National	Male
03	Social advocate and founder of NGO to advise families on support services and companies to promote inclusive employment.	UAE	Recovered from brain injury as child. University degree in international relations.	Country national	Male
04	Member of Global Youth Council set up by international NGO and representing African region.	Sub Sharan Africa	In my role I am representing my fellow colleagues with intellectual disability. And locally, back home, I am a sports assistant. I’m a health messenger and also the vice chairperson for the local athlete input council.	Regional National	Male
05	Senior social media editor in print and radio.	UAE	A wheelchair user, I use my current role to shape or reframe the media, pushing more inclusion and making the media more diverse.	Country National	Female
06	Trainer of leaders in sports and self-employment.	Sub- Sharan Africa	At my school I studied automotive engineering. I’ve been able to mentor around 20 students in my campus. We have an organisation there that we are running of helping young people to start their own businesses.	Regional National	Male
07	A psychologist and university teacher and researcher. A mother of a child with autism.	Zambia	To engage with parents and communities and to encourage them that there’s something they can do about it.	Country National	Female
08	Volunteer mentor to young adults with intellectual disability.	Zimbabwe	A social worker with over 10 years of practice and a volunteer for the Lions Club International with a local sports organisation. Former cricketer.	Country National	Male
09	National Sports Director for an organisation serving people with intellectual disabilities.	Pakistan	I have been working in my role since 2019 and before that as a sports development volunteer with the organisation for whom I work. I have had a strong passion for sports, for community development, leadership and they all come together in my present job.	Country National	Female
10	Country Director employed by an international company/contractor working in-country on inclusion.	Uzbek- istan	Adjunct university faculty member on doctoral Education Leadership Program. Previously worked in Central Asia, South Caucasus, Eastern Europe and Southeast Asia on inclusive education projects.	Expatriate to the Country	Female
11	National co-ordinator of inclusion project led by sports NGO.	Argentina	Qualified as psychopedagogue. Preparing training courses and guides for implementers of inclusion in schools across the county.	Country National	Female
12	Project co-ordinator of youth projects in our national sports NGO.	El Salvador	I have experienced disability since birth through my uncle (who lives with an intellectual disability). My experience has made me passionate about advocating for human rights and constantly seeking new knowledge to change the world.	Country National	Female
13	Director for global development and government relations in a regional NGO.	Sub- Sharan Africa	I have been with the NGO for 17 years, having started as a volunteer, then paid manager, national director in one country and as a regional director for 5 years.	Regional Director	Female
14	Regional Director of an international NGO responsible for planning, monitoring data, and analytics.	South Asia	University graduate, with over 15 years experience with previous positions at HQ.	Expatriate to Region	Male
15	Regional Officer for inclusion with an international sports NGO.	Latin America	A university graduate with responsibility for inclusion in schools and promoting the leadership of people with intellectual disabilities.	Regional/ National	Male
16	Chief inspirational officer and Board Member of an international NGO for sports and people with intellectual disability.	Global	A person with intellectual disability whose been a sports participant and advocate since 1970.	US Citizen	Female

## Data Availability

Anonymised transcripts of the interview are available from the author but subject to his obtaining permission from the interviewee to share them.

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
