# Peer review of "Nurturing Leaders in Community-Based, Primary Healthcare Services for People with Disabilities in Low- and Middle-Income Countries"

_ijerph, 2025, doi:10.3390/ijerph22040622_

Round 1
Reviewer 1 Report
Comments and Suggestions for Authors
-
Strengthen the Introduction
- Adding quantitative data on healthcare access for disabled individuals in LMICs would provide more context.
- A brief mention of previous studies on leadership in community-based care would enhance the literature review.
-
Clarify Methodology
- Further detail on the manual validation process of AI-generated themes would improve methodological transparency.
- Address potential biases of AI in qualitative research.
-
Enhance Discussion on Leadership Models
- While the study discusses transformative and servant leadership, a clearer comparison of these models with existing healthcare leadership styles in LMICs would add depth.
-
Improve Writing Clarity
- Minor grammatical improvements and restructuring of sentences could enhance readability.
Improve Writing Clarity
-
- Minor grammatical improvements and restructuring of sentences could enhance readability.
Author Response
Please see the attachment which contains responses to the comments for all four reviewers.

Reviewer 2 Report
Comments and Suggestions for Authors
Abstract
The abstract states that the study aims to explore "the necessary leadership qualities" but does not elaborate on the expected impact of the study.
It does not mention the type of study. It is advisable to include a brief description of the study design.
It does not specify how the participants were selected.
Introduction
The introduction mentions that the study seeks to understand leadership qualities but does not explicitly formulate a research question.
It refers to some studies but does not clearly establish the gap in the literature that the study intends to address. What is the significance of this study? What new insights does it offer?
The article asserts that community leadership is essential but does not discuss why this topic needs to be explored empirically.
Methodology
The article includes only 16 participants but does not explain why this number was considered sufficient.
The study states that participants were selected using purposive sampling, but it does not clarify the specific criteria used for their selection.
The article mentions that the analysis was conducted with AI support (Microsoft Co-Pilot) but does not clearly describe the analytical process. How were the themes extracted? Was data triangulation performed?
The article states that the study followed the Declaration of Helsinki, but there is no mention of approval from an ethics committee.
Results
The article presents the results in a highly descriptive manner, without discussing differences among participants or the context of the findings.
Even though this is a qualitative study, it would be useful to include some basic statistics (e.g., how many participants mentioned each theme?).
Discussion
The article suggests that the identified leadership qualities are universal but does not discuss potential cultural or contextual variations.
Conclusion
There is no clear statement on the impact of the findings. Therefore, it is recommended to present more concrete implications of the results.
There is a lack of recommendations for future research. The article could suggest longitudinal studies or research with a larger sample size.
References
The references do not comply with the journal's formatting guidelines. It is recommended to review and correct them accordingly.
Author Response
Please see the attachment which contains responses to the comments for Reviewer 2

Reviewer 3 Report
Comments and Suggestions for Authors
- Strengthen the rationale for the study in the introduction.
- Add more detailed explanations of the methods.
- Improve the description of the results to make them easier to understand.
- Revise the English language for clarity.
- Cited references appear appropriate and relevant. However, it would be good to ensure that all citations are up to date.
The introduction provides a good context for the study and presents relevant references. However, it may be useful to reinforce the rationale for the study by highlighting specific gaps in the literature. The research design seems appropriate for the proposed objectives. If there is room for improvement, a more detailed explanation of the choice of method could be included. The methods are well described, but it would be useful to clarify some details, such as the exact criteria for data selection or the justification for certain approaches.
The results are presented clearly, but could be improved with additional descriptions to facilitate reader comprehension. The conclusions are consistent with the results presented. However, it would be useful to include more direct suggestions about the practical implications of the findings.
Given the proposed objective of this article, we believe that it would be more appropriate for the figures and tables to be adjusted to the journal's demands. It is necessary to check that all references cited throughout the article are in the correct format, as some variations were identified.
We recommend that the English be revised to improve the clarity of the article.
Author Response
Please see the attachment which contains responses to the comments for Reviewer 3

Reviewer 4 Report
Comments and Suggestions for Authors
Dear Author,
I appreciate the opportunity to review your manuscript. The topic addressed is relevant and covers fundamental aspects of quality and leadership skills, as well as ways to enhance leadership in community services for people with disabilities.
Below, I present some suggestions to improve the quality of your manuscript.
1. Introduction
The introduction has a clear and logical structure, well-organized and well-founded. However, some points can be improved:
Line 85: Clarify the meaning of the acronym NGO.
Lines 99-108: The citation needs to be properly referenced, as it is unclear where the author obtained this information. The reader only identifies the source when accessing reference [21].
Section 1.1: I suggest removing this subdivision and integrating its content into the introduction to maintain a more natural flow.
Lines 126-129: This excerpt belongs to the Methods section, as it discusses participant selection.
Lines 129-132: The presented questions should be reformulated to clearly reflect the research questions derived from the study’s objectives. Currently, they appear to be the questions asked to participants.
2. Methods
I suggest some improvements to the Methodology section to strengthen the study’s foundation and transparency, particularly in justifying the methodological approach, discussing ethical aspects, and describing AI usage. Some specific recommendations:
The author mentions that the study follows a qualitative approach. It would be important to specify which type of qualitative study was used.
The Methods section should detail the research process, ensuring its validity and reproducibility.
Lines 146-147: The mentioned limitation should be moved to the Limitations section.
There should be a section with well-defined inclusion and exclusion criteria. The author clarifies the time frame in which the interviews were conducted. However, specifying the exact year and month would be beneficial, as time factors can influence the study’s findings.
2.1. Sample
The description/characterization of study participants should be included in the Results section, forming its first part. I emphasize that the Methods section is meant for describing the research process.
It would be beneficial to clearly specify the month and year of the interviews, as certain temporal factors may influence the results.
I suggest organizing the Methods section as follows (this would make the entire process clearer and more organized for the reader):
2.1. Participants (including inclusion and exclusion criteria)
2.2. Procedures for participant recruitment
2.3. Data collection instruments
2.4. Data collection procedure
2.5. Qualitative analysis using AI
2.6. Data analysis
2.7. Ethical procedures
3. Results
I suggest beginning this section with the characterization of participants, including Table 1 for better visualization.
Lines 410-414: These excerpts contain data interpretation, which is more appropriate for the Discussion section.
4. Discussion
Lines 471-472: Clearly specify which study the discussion refers to.
Section 4.3 (AI as a research and evaluation tool): The discussion should focus on the obtained results rather than the methods. Justifications for AI usage could be moved to the Methods section.
Section 4.4 (Limitations): I suggest that this section exclusively contain the study’s limitations, while suggestions for future research could be included in the Conclusion.
5. Conclusion
Consider numbering this section as "5. Conclusion."
Line 561: Since this is a qualitative study based on a relatively small sample, claiming that it validates the universality of the results may be an overstatement. I suggest rephrasing this statement.
My suggestions aim to enhance the clarity, structure, and foundation of the manuscript, contributing to its overall quality. I hope these recommendations are useful for improving your work.
Best regards,
Author Response
Please see the attachment which contains responses to the comments for Reviewer 4

Round 2
Reviewer 2 Report
Comments and Suggestions for Authors
I consider that the suggested modifications were appropriate to accept the present article for publication
Author Response
I consider that the suggested modifications were appropriate to accept the present article for publication.
Response: Many thanks for this confirmation - much appreciated.
Reviewer 4 Report
Comments and Suggestions for Authors
Dear Author,
Again, Thank you for your valuable scientific contribution and for introducing new methodologies into the research domain, mainly through artificial intelligence. Innovation is essential, and it is important to recognize the potential of such approaches within qualitative inquiry.
The manuscript has shown significant improvements, and I appreciate your attention to my suggestions as well as those of the other reviewers.
I agree that describing the sample is a prerequisite for presenting results in qualitative research. However, I would like to emphasize that the demographic characterization of participants is typically presented in the Results section, usually at the beginning, as an introductory paragraph or subsection. In qualitative studies, this information is considered part of the data, helping contextualize and interpret participants’ narratives. As such, it is not usually treated as part of the “Methods” section, but rather as a component of the results, providing readers with a clearer understanding of who the participants are.
For these reasons, I recommended its inclusion in the Results section, a position I continue to support.
Once again, thank you for your scientific contribution.
Best regards,
Author Response
Again, Thank you for your valuable scientific contribution and for introducing new methodologies into the research domain, mainly through artificial intelligence. Innovation is essential, and it is important to recognize the potential of such approaches within qualitative inquiry. The manuscript has shown significant improvements, and I appreciate your attention to my suggestions as well as those of the other reviewers.
I agree that describing the sample is a prerequisite for presenting results in qualitative research. However, I would like to emphasize that the demographic characterization of participants is typically presented in the Results section, usually at the beginning, as an introductory paragraph or subsection. In qualitative studies, this information is considered part of the data, helping contextualize and interpret participants’ narratives. As such, it is not usually treated as part of the “Methods” section, but rather as a component of the results, providing readers with a clearer understanding of who the participants are. For these reasons, I recommended its inclusion in the Results section, a position I continue to support. Once again, thank you for your scientific contribution.
Response: Many thanks for your kind comments - they are much appreciated. I have moved the description of the participants to the results section.